# ADAM10 and ADAM17, Major Regulators of Chronic Kidney Disease Induced Atherosclerosis?

**DOI:** 10.3390/ijms24087309

**Published:** 2023-04-15

**Authors:** Sanne L. Maas, Marjo M. P. C. Donners, Emiel P. C. van der Vorst

**Affiliations:** 1Institute for Molecular Cardiovascular Research (IMCAR), RWTH Aachen University, 52074 Aachen, Germany; 2Aachen-Maastricht Institute for CardioRenal Disease (AMICARE), RWTH Aachen University, 52074 Aachen, Germany; 3Department of Pathology, Cardiovascular Research Institute Maastricht (CARIM), Maastricht University Medical Centre, 6229 ER Maastricht, The Netherlands; marjo.donners@maastrichtuniversity.nl; 4Interdisciplinary Center for Clinical Research (IZKF), RWTH Aachen University, 52074 Aachen, Germany; 5Institute for Cardiovascular Prevention (IPEK), Ludwig-Maximilians-University Munich (LMU), 80336 Munich, Germany

**Keywords:** a disintegrin and metalloprotease, chronic kidney disease, cardiovascular diseases, atherosclerosis

## Abstract

Chronic kidney disease (CKD) is a major health problem, affecting millions of people worldwide, in particular hypertensive and diabetic patients. CKD patients suffer from significantly increased cardiovascular disease (CVD) morbidity and mortality, mainly due to accelerated atherosclerosis development. Indeed, CKD not only affects the kidneys, in which injury and maladaptive repair processes lead to local inflammation and fibrosis, but also causes systemic inflammation and altered mineral bone metabolism leading to vascular dysfunction, calcification, and thus, accelerated atherosclerosis. Although CKD and CVD individually have been extensively studied, relatively little research has studied the link between both diseases. This narrative review focuses on the role of a disintegrin and metalloproteases (ADAM) 10 and ADAM17 in CKD and CVD and will for the first time shed light on their role in CKD-induced CVD. By cleaving cell surface molecules, these enzymes regulate not only cellular sensitivity to their micro-environment (in case of receptor cleavage), but also release soluble ectodomains that can exert agonistic or antagonistic functions, both locally and systemically. Although the cell-specific roles of ADAM10 and ADAM17 in CVD, and to a lesser extent in CKD, have been explored, their impact on CKD-induced CVD is likely, yet remains to be elucidated.

## 1. Introduction

Chronic kidney disease (CKD) is a complex disease and evolves from numerous risk factors which lead to the alteration of the structure and function of the kidney. These alterations are irreversible and can progress asymptomatically over months or even years, in which the kidneys slowly and progressively lose their ability to filter waste products and excess fluids from the blood, leading to a buildup of (uremic) toxins in the body. The diagnosis of CKD relies on the identification of such structural damage to the kidney and the detection of a chronic reduction of kidney function. The main indicator of kidney function is the glomerular filtration rate (GFR), which equals the total fluid filtered through nephrons per unit of time [1]. After irreversible nephron loss, CKD patients can reach end-stage renal disease (ESRD), also known as kidney failure. This stage especially goes hand in hand with a massive impact on quality of life as these patients require frequent dialysis or even kidney transplantation. CKD was ranked tenth on the list of global causes of death in 2019, accounting for 12.2 deaths per 100,000 people [2]. This number is expected to reach 14 per 100,000 people in 2030 [2]. Furthermore, CKD is also associated with substantial morbidity as 35.8 million disability-adjusted life-years were lost worldwide in 2017 [3]. This increasing incidence and prevalence of (advanced) CKD is accounted for by, among others, aging populations, hypertension, obesity, and the increasing prevalence of type 2 diabetes [4] (Table 1). Additionally, at the moment, many patients are only diagnosed in the later stages of the disease, leading to a delayed administration and start of therapeutic strategies [5,6,7]. CKD is also an important risk factor for other morbidities, such as cardiovascular diseases (CVD) and the related mortality. Both CKD and CVD are major public health problems that affect millions of people worldwide [8,9,10] and can lead to life-threatening complications if left untreated. Although both conditions are widely studied independently from each other, interestingly, there seems to be a strong link between these two conditions, as CKD patients are at a remarkably increased risk of developing CVD [11,12,13]. Strikingly, a cardiovascular (CV) event, rather than ESRD, is the leading cause of death in this high-risk population of CKD patients [11,14,15]. Where 1.2 million people died from CKD, an additional 1.4 million CVD-related deaths could be attributed to impaired kidney function [3]. In fact, the prevalence of CVD in CKD patients is greater than in the general population in all disease stages at any age, but a significant increase could only be observed in more advanced stages of the disease (CKD stage 3–5) [11,16]. CKD is therefore recognized as a major risk factor for CVD, which is interestingly rather independent of other conventional CVD risk factors [17]. Classical risk factors for CV morbidity and mortality in the general population, such as age, gender, hypertension, smoking, diabetes mellitus, and dyslipidemia, did not fully predict outcomes in CKD patients [18], suggesting additional CKD-related pathogenic processes may be at play, as also highlighted in Table 1.

### 1.1. Pathogenesis and the Role of Cellular Cross-Talk

Early-stage CKD is characterized by the reduction of microvascular density (microvascular rarefaction) and tubular atrophy, which are both hallmarks that lead to tissue hypoxia, inflammation, and fibrosis and thereby drive CKD progression [23]. The exact pathophysiological mechanisms underlying microvascular rarefaction are still rather elusive. Therefore, a better understanding is needed to enable early disease detection and therapeutic intervention. Upon kidney damage, albuminuria will develop, which is associated with systemic capillary rarefaction and endothelial dysfunction [24]. Endothelial dysfunction can additionally be induced by a large variety of stimuli, such as growth factors (e.g., TGFβ, VEGF, EGF), pro-inflammatory molecules such as TNF and IL-6 [25], hyperlipidemia, but also uremic toxins, which are mainly present at later stages of CKD. Dysfunction of the endothelium not only leads to increased endothelial inflammation and permeability but also stimulates endothelial-to-mesenchymal transition (EndoMT), which leads to capillary loss and fuels renal fibrosis [26]. Similarly, structural and functional changes in tubular epithelial cells (TECs) upon kidney damage lead to (partial) epithelial-to-mesenchymal transition (EMT), thereby driving inflammation as well as fibrosis and bolstering the progression of CKD. It is generally accepted that there is extensive cross-talk between both kidney TECs and peritubular endothelial cells [27], as well as between kidney endothelium/epithelium and the systemic vasculature. However, the relative contribution of both cell types is still largely unknown. Furthermore, in the renal tubules, several stimuli influence TECs, which are major components of the tubules in the kidney. TECs are vulnerable to several deleterious factors such as hypoxia, toxins, and proteinuria. Upon kidney injury, TECs activate several repair mechanisms. However, maladaptive repair, such as pericyte dissociation from the endothelium, pericyte proliferation and differentiation, lead to an increase of myofibroblasts resulting in the progressive deposition of collagen I, persisting presence of M1 macrophages, and G2/M arrest of tubular cells, driving interstitial inflammation and fibrosis which eventually might result in CKD [28].

Besides local intercellular cross-talk, CKD-associated factors (e.g., hormones, enzymes, cytokines, but also uremic toxins) mediate systemic cross-talk, affecting the systemic vasculature, contributing to accelerated atherosclerosis, vascular calcification, and related CVD complications. Beyond the traditional risk factors, some additional mechanisms are assumed to contribute to the progression of CVD in CKD patients. Firstly, oxidative stress plays a key role in the development of atherosclerosis by the production of bioactive molecules, modification of low-density-lipoprotein (LDL), consumption of nitric oxide resulting in endothelial dysfunction, and production of various oxidative reactants and diffusible free radical species [29,30,31]. Captivatingly, a significant association between the baseline level of oxidative stress and the incidence of CVD has been observed in CKD patients [32]. Secondly, hormones [33,34,35,36], enzymes, and cytokines [37,38,39] are released in response to kidney injury and/or kidney damage, resulting in alterations of the vasculature. Thirdly, CKD-associated mediators, e.g., uremic toxins [40], as well as hemodynamic alterations contribute to vascular damage [41]. However, many of the exact underlying mechanisms behind the highly increased CVD risk in CKD patients still remain elusive and therefore are an active field of research.

### 1.2. ADAMs

The underlying processes of CKD and CKD-induced CVD are rather diverse and involve a wide variety of signaling pathways; however, shedding is a major way of commutation in the cellular cross-talk between these pathologies. Over the past few decades, many studies in the context of CVD and kidney homeostasis have focused on a disintegrin and metalloproteinases (ADAMs), which are constitutively expressed zinc-dependent membrane proteases. Interestingly, these studies demonstrated that ADAMs play a diverse role and control developmental processes, tissue remodeling, inflammatory responses, as well as proliferative signaling pathways by either modifying or shedding proteins (reviewed in [42,43,44,45,46,47,48,49,50]). In humans, 21 ADAM family members have been identified so far [51], although only 12 ADAMs (ADAM8–10, 12, 15, 17, 19–21, 28, 30, and 33) actually display proteolytic activity [52]. Additionally, the expression pattern of the different ADAMs is highly variable, although ADAM10 and ADAM17 are expressed at a substantial basal level in almost all cell types. Additionally, ADAM expression is regulated by endogenous proteins known as tissue inhibitors of metalloproteases (TIMPs) [50]. TIMPs are well known to also regulate matrix metalloproteases (MMPs) [53]. Interestingly, there seems to be a link between ADAMs and MMPs as, for example, ADAM-mediated release of TNF-α induces MMPs (MMP-2, -8, -9) via a positive feedback loop [54,55,56]. Furthermore, the expression of ADAMs can be induced by different stimuli. For example, ADAM10-dependent shedding can be induced by intracellular calcium signaling, whereas ADAM17 activity is induced by protein kinase C [57]. ADAMs are synthesized as inactive precursors (also known as proenzymes), consisting of seven domains. The prodomain (pro) is coupled to the metalloprotease domain (MP), after which ADAMs harbor a disintegrin domain (D), followed by a cysteine-rich domain (C), an EGF-like domain (EGF; except for ADAM10 and ADAM17), a transmembrane domain (TD), and finally a short cytoplasmic domain (CD) [25] (Figure 1). In order to convert these precursors into active ADAM proteases, enzymatic removal of the N-terminal prodomain has to take place.

The expression and function of ADAMs in the kidney are heterogeneous and dependent on many variables, such as the underlying disease, the disease stage, the localization of the proteases as well as their regulators in the kidney. The two most-studied family members of the ADAM family are ADAM10 and ADAM17 [43,58], which are highly expressed on the cell surface of epithelial cells in the distal tubule [59] as well as endothelial cells (ECs), particularly in diseased endothelium [60]. Their main function is the cleavage of a variety of membrane-bound proteins, including (receptors for) many cytokines and growth factors, thereby shedding soluble proteins. This shedding process represents a major mechanism by which ADAM10 and ADAM17 influence cellular responses in the kidney itself, but also the intra-renal as well as systemic intercellular communication [61]. ADAM10 and ADAM17 shed partly overlapping substrates, although they appear to have opposite functions in atherosclerosis [62,63]. Interestingly, only a limited amount of literature is available regarding the role of ADAM10 in renal disease, and there is especially a lack of studies that investigate the role of these ADAMs in CKD-induced atherosclerosis, leaving these ADAMs and their substrates as an interesting target to study in this context. Several key pathophysiological processes have been identified in which ADAM10 and ADAM17 play a key role (Table 2). The most important ones will be discussed in the following chapters. Therefore, this narrative review will discuss the current state of the art regarding the role of ADAM10 and ADAM17 in CKD and CVD and will for the first time shed light on the involvement of these ADAMs in CKD-induced CVD.

## 2. Role of ADAM10/17 in CKD

It has been recognized that dysregulation of ADAMs occurs in CKD, and genetic targeting of ADAMs in different mouse models of kidney disease showed that they can have detrimental and protective roles. In particular, substrate shedding by ADAM10 and ADAM17 leads to a large variety of consequences for the structure and functionality of the kidney (Table 2). The main substrates, and the biological effect of its shedding in the context of CKD, will be discussed below.

### 2.1. ADAM17 and Its Role in CKD

In human kidneys of healthy individuals, *ADAM17* is especially expressed on a genetic level in the distal tubules, as shown by RNA in situ hybridization, whereas no *ADAM17* gene expression is observed in the glomerular endothelium, glomerular mesangium, peritubular capillaries, and proximal tubules; a moderate expression is observed in the glomerular parietal cells and podocytes; and a moderate to strong expression is observed in renal arterial ECs and smooth muscle cells (SMCs) [59,152]. However, in CKD patients, the mRNA expression of *ADAM17* is upregulated in the glomerular parietal epithelium and podocytes, and a de novo expression is observed in the glomerular endothelium and mesangium, peritubular capillaries, and proximal tubules in the kidneys compared to healthy individuals, indicating that disease development has a major influence on *ADAM17* expression [59,152]. Studies focusing on ADAM17 in kidney biopsies from patients with CKD and in experimental mouse models of renal disease suggest the importance of ADAM17 in kidney inflammation, fibrosis, and disease progression [45,61]. For example, the NEFRONA study showed an increase in circulating ADAM17 in plasma samples from patients with CKD (without a previous history of CVD) with advanced CKD (stage 5D) compared to patients with moderate CKD (stage 3–5) [153]. Circulating ADAM17 has furthermore been recognized as an independent marker to identify CV events in CKD patients, although circulating ADAM activity (especially ADAM17) as a risk factor for CKD progression was only found in male patients [153].

The role of epidermal growth factor receptor (EGFR), a member of the ErbB family of tyrosine kinase receptors [46], signaling in kidney pathology has already been thoroughly investigated. Captivatingly, EGFR ligands are shed by ADAM17 as well as ADAM10, and whereas amphiregulin (AREG) and transforming growth factor α (TGFα) are predominantly shed by ADAM17, betacellulin and EGF are shed by ADAM10 [64] (Figure 2). Interestingly, the EGFR ligands activate the EGFR signaling pathway, which can in turn activates ADAM17 or ADAM10, thereby generating a positive feedback loop. In CKD patients, AREG is highly upregulated in the kidneys [61]. In vitro experiments suggest that *ADAM17* is upregulated by AREG and that a positive feedback loop results in the ability of AREG to upregulate itself [61]. Furthermore, the shedding of AREG increases EGFR activation. This leads to the enhanced production of pro-fibrotic and pro-inflammatory cytokines in primary murine tubular cells isolated from injured kidneys and in a human proximal tubular cell line in vitro [61,154]. Furthermore, in vivo data showed that *Adam17* hypomorphic mice, specific *Adam17* inhibitor-treated WT mice, or mice with an inducible knockout of *Adam17* in the proximal tubule (*Slc34a1*-Cre) were significantly protected from kidney fibrosis [61]. Collectively, these findings depict that AREG can have pro-inflammatory and pro-fibrotic effects. Moreover, accumulating evidence shows that ADAM17 might regulate tissue remodeling and inflammation in the kidney via EGFR transactivation [47,155]. ADAM17 shedding of TGFα also leads to the activation of the EGFR–MAPK pathway [153]. Activation of this pathway results in the stabilization of ADAM17, which induces a devastating feed-forward loop, resulting in the progression of kidney dysfunction [156]. Elevated ADAM17 expression in the kidneys of CKD patients co-localized with TGFα in the fibrotic regions [59]. Furthermore, ADAM17 releases the EGFR ligand HB-EGF, which also results in the activation of EFGR signaling, leading to the upregulation of pro-inflammatory factors as well as increased inflammatory cell infiltration [59] (Figure 2).

An in vitro study with TECs showed aldosterone-induced upregulation of pro-inflammatory genes as well as overexpression of pro-inflammatory factors upon ADAM17/EGFR activation. In line with this, blockage of the ADAM17/EGFR pathway had a counteracting effect in response to aldosterone and resulted in an anti-inflammatory environment [157]. Furthermore, high glucose conditions promoted TEC injury due to increased oxidative stress, which was prevented by the inhibition of ADAM17, suggesting that ADAM17 is an important mediator of inflammatory processes in the renal tubule [91]. Moreover, ADAM17 pathway activation leads to TEC proliferation and EMT, which is defined by an amplified expression of cellular collagen and fibronectin [61,158]. The sustained activation of EGFR signaling by ADAM17 also induces interstitial fibrosis due to an increased release of pro-fibrotic factors [61]. These findings reflect the complexity of the regulatory network between TECs and ADAM17 in cellular processes such as inflammation and fibrosis.

In addition to EGFR ligands, the membrane-bound chemokines CX3CL1 [102,103,104,105] as well as CXCL16 [104,105,108] are shed in the kidney by ADAM17, but also by ADAM10 (Figure 2). Upregulation of both chemokines in mice resulted in the recruitment of monocytes, NK cells, and T cells, which has been implicated in the pathogenesis of inflammatory and fibrotic kidney diseases [159,160]. However, shedding can alter the function of chemokines due to the conversion of a membrane-bound chemokine with cellular adhesion function to a chemoattractant as a soluble variant. ADAM17 and ADAM10 might thus support the local fine-tuning of cell recruitment facilitated by CX3CL1 and CXCL16 [103,104,105]. However, whether CX3CL1 or CXCL16 shedding encourages or diminishes kidney damage remains unclear.

### 2.2. ADAM10 and Its Role in CKD

It has been demonstrated that ADAM10 plays a crucial role during the development of the kidney by mediating Notch-induced effects [161,162,163]. For example, ADAM10-mediated Notch signaling is vital for the development of the kidney vasculature, especially promoting the development as well as the maturation of the glomerular endothelium [162,163]. Furthermore, in vitro studies showed that a knockdown of *Adam10* disrupted proximal tubule development [164]. Interestingly, several embryonic pathways, e.g., Notch, Wnt, and Hedgehog signaling are revived upon kidney injury [165,166].

An overview of some major substrates that are shed by ADAM10 can be found in Table 2. Interestingly, ADAM10 is upregulated in the kidney tissue of CKD patients, clearly suggesting that it plays a key role in disease development [167]. Besides regulating cellular sensitivity to environmental stimuli, ADAM10 also orchestrates the release of soluble mediators with (ant)agonistic, paracrine functions involved in inflammation, cell survival/proliferation, and fibrosis. Conceivably, these factors will mediate intra-renal cellular cross-talk between epithelial cells and the endothelium of the peritubular capillaries [27].

In a rat model of CKD, increased expression of *Adam10* contributed to EMT of tubular epithelia and increased kidney fibrosis [167]. Moreover, in a mouse model, it could be demonstrated that activation of ADAM10 promoted kidney interstitial fibrosis and eventually renal dysfunction [164]. Furthermore, previous in vivo studies reported that endothelial-specific *Adam10*-deficient mice show vascular abnormalities in the kidney, such as enlarged and hypercellular glomeruli with occasionally dilated peripheral capillaries, as well as an increased amount of mesangial collagenous matrix in glomeruli [163].

The shedding of additional transmembrane proteins has also been shown to influence the morphology of the kidney, for example, cleavage of CXCL16 by ADAM10 led to lupus nephritis and acute tubular necrosis [107,108], whereas IL6-R shedding by ADAM10/17 resulted in acute crescentic glomerulonephritis and lupus nephritis [114], and Notch shedding by ADAM10/17 led to renal fibrosis and glomerulosclerosis [164,168]. Additionally, the shedding of TNF-α and TGF-β by ADAM17 resulted in increased fibrosis, glomerulosclerosis, inflammation, protein matrix accumulation, and neutrophil and macrophage infiltration by both the TNF receptor (TNFR) and EGF receptor (EGFR) signaling pathways [47,61,147]. Furthermore, it has been shown that substrates of ADAM10/17 such as Meprin influence kidney function as the Meprin inhibitor, actinonin, lowered blood urea nitrogen and serum creatinine levels in the presence of renal sepsis [131].

## 3. Role of ADAM10/17 in Atherosclerosis

### 3.1. Substrate Cleavage by ADAM10/17 in the Context of Inflammation

Besides a crucial role in CKD, ADAM10 and ADAM17 have been more elaborately studied in the context of CVD, particularly in atherosclerosis, the main underlying cause of CVD [43,57,169]. Atherosclerosis is a multifactorial lipid-driven inflammatory disease in which various cell types and multiple crucial mediators play an important role. Various mediators have already been identified as substrates for ADAM10 and/or ADAM17 (extensively reviewed in [57,169] and listed in Table 2). The main mediators during inflammation and atherogenesis are the chemokines that play an essential role in the recruitment of leukocytes to the injured vascular wall. The membrane-bound chemokines CXCL16 and CX3CL1 are not only shed by ADAM10/ADAM17 in the kidney but also play an important role in the context of atherosclerosis [104,106]. Increased cleavage of these chemotactic proteins would on the one hand result in an increased attraction of leukocytes to the vessel wall and on the other hand, diminish the adhesive capacity of the vessel wall, resulting in opposing effects in relation to inflammation and atherogenesis [104,106]. Additionally, in this leukocyte recruitment process, the shedding of intercellular adhesion molecule 1 (ICAM-1) and vascular cell adhesion molecule 1 (VCAM-1) by ADAM17 plays an important role as these adhesion molecules are crucial for the adherence of leukocytes to the vessel wall [98,118]. Once leukocytes adhere to the vessel wall, they still have to cross the endothelium in a process called transmigration. In this context, the junctional molecules vascular endothelial (VE)-cadherin and junctional adhesion molecule A (JAM-A) play an important role as they maintain the integrity of the endothelial layer [74,119,170]. It was shown that ADAM10/17 can cleave these junctional molecules, resulting in increased vascular permeability, thereby promoting leukocyte transmigration [74,119,170,171]. Based on the important role of ADAM10 and ADAM17 in inflammation and the related leukocyte recruitment studied mainly in in vitro settings, it is highly likely that these ADAMs also play a crucial role in atherosclerosis formation in vivo.

### 3.2. ADAM17 as a Mediator of Atherosclerosis

ADAM17 expression has indeed been associated with atherosclerosis development. For example, quantitative trait locus mapping in mice demonstrated that increased *Adam17* expression is associated with atherosclerosis resistance [172]. On the other hand, it was shown that *Adam17* is expressed in murine atherosclerotic lesions and expression increased with lesion progression, with a concomitant increase in plasma levels of soluble TNFRs [173]. In addition to these murine data, *ADAM17* expression was also upregulated in advanced human atherosclerotic lesions [174], and a positive association between circulating levels of ADAM17 substrates (soluble (s)ICAM-1, sVCAM-1, sIL6R, and sTNFR1) and the risk for a second major CV event was also observed in humans [175]. Furthermore, ADAM17 expression was predominantly associated with CD68-positive cells of monocytic origin, although especially in carotid artery atherosclerotic plaques, ADAM17 expression was also observed in vascular cells [174].

Although these association studies showed somewhat contradictory results, a more recent study by Nicolaou et al. investigated the causal role of ADAM17 in atherosclerosis development. Since full-body *Adam17*-deficient mice are, similar to *Adam10*-deficient mice, embryonically lethal, they used *Adam17* hypomorphic mice (*Adam17^ex^*^/ex^) [176] that express very low levels of ADAM17. This study showed that ADAM17 has an atheroprotective role in atherosclerosis development as *Adam17* hypomorphic mice have atherosclerotic lesions that are 1.5-fold larger compared to controls [177]. The proposed mechanism was the reduced shedding of membrane-bound TNF and TNF-receptor 2 (TNFR2) in mice with very low ADAM17 expression, resulting in constitutive activation of TNFR2-signalling [177]. Consequently, several cellular functions are also disturbed upon *Adam17*-deficiency, such as increased proliferation and reduced apoptosis in macrophages and SMCs and an increased adhesion of macrophages to ECs in vitro, which are all atherosclerosis-promoting effects [177]. Because these effects were alleviated with the knockdown of *Tnfr2*, it was confirmed that TNF-TNFR2 signaling is a crucial mechanism behind the observed ADAM17-mediated effects on atherosclerosis [177].

The full-body approach of the study by Nicolaou et al. did not allow for the determination of which exact cell type is responsible for the observed effects. Using the Cre-flox system, cell-specific effects of ADAM17 in atherosclerosis could be distinguished [62]. Myeloid-specific *Adam17*-deficiency (*LysM*-Cre driven) resulted in an almost 2-fold increase in atherosclerotic lesion sizes with a more advanced lesion phenotype, characterized by reduced relative macrophage content with increased relative SMC and collagen content [62]. Although it is plausible to speculate that the interference with TNFR2 signaling is also the underlying mechanism in this mouse model, perhaps together with the interference of MMP expression (similar to the myeloid-specific *Adam10-*-deficient model, as described below), such underlying mechanisms have not been studied yet and thereby remain an active field of research. Interestingly, endothelial-specific *Adam17*-deficiency (*Bmx*-Cre driven) demonstrated the opposite effects on lesion development, as the lack of endothelial ADAM17 resulted in significantly reduced atherosclerotic lesion sizes [62].

The notion that myeloid and endothelial ADAM17 have opposing effects on atherosclerosis formation is also of crucial importance for potential future therapeutic options to target ADAM17. So far, many studies in other diseases, e.g., cancer, have focused on the general inhibition of ADAM17, which at least in the context of atherosclerosis would probably not be effective due to the contradicting cell-specific effects and may even lead to unwanted side-effects. Therefore, cell-specific or substrate-specific targeting is needed.

### 3.3. ADAM10 as a Mediator of Atherosclerosis

Several studies have already highlighted that ADAM10 is associated with atherosclerosis development in humans. For example, we previously showed that ADAM10 expression is significantly increased during atherosclerotic plaque progression compared to healthy human vessels and early atherosclerotic lesions [74]. Furthermore, a polymorphism in the *ADAM10* promoter (rs653765) which results in an increased ADAM10 expression was found to be associated with atherosclerotic cerebral infarction [178]. However, such associations do not show the cause of the effect, which needs to be explored in animal models. However, a limiting factor in the evaluation of this causal role of ADAM10 in atherosclerosis in vivo is the fact that full-body *Adam10*-deficient mice are embryonically lethal [179], making it impossible to investigate the global effect of ADAM10. However, cell-specific models have so far proven to be highly useful to at least elucidate the role of cell-specific ADAM10 in atherosclerosis formation.

Using a conditional knockout model in which mice lack ADAM10 specifically in myeloid cells (*LysM*-Cre driven), we could, for example, elucidate the causal role of myeloid ADAM10 on atherosclerosis formation using bone marrow transplantation into atherosclerosis-prone *Ldlr^−/−^* mice [180]. Although lesion size was not affected by the lack of myeloid ADAM10, lesions demonstrated an increased collagen content [180], indicating that myeloid ADAM10 affects the fibrotic processes in atherosclerotic lesions. In line with this, it was shown in vitro that macrophages lacking ADAM10 have a significantly reduced expression of MMP9 and 13 and a reduced activity of MMP2. It has previously been shown that ADAM10 has the direct matrix-degrading capacity, although this observation was made in a rather artificial in vitro setting [181]. Additionally, macrophages lacking ADAM10 also displayed a more anti-inflammatory (M2-like) phenotype, characterized by increased IL-10 secretion and reduced TNF and IL-12 release, although there was no change in profibrotic M2 markers, such as arginase 1 or TGF-β. Overall, ADAM10 modulates fibrosis by reducing MMP activity either directly or indirectly.

Contrasting this role of myeloid ADAM10 in atherosclerosis formation, endothelial *Adam10*-deficiency resulted in a markedly increased atherosclerotic lesion size, which coincided with an increased necrotic core and reduced macrophage content, though lesional collagen content was unaffected [63]. Interestingly, a majority of plaques from mice that lacked endothelial ADAM10 showed features of intraplaque hemorrhage and neovascularization, which is normally rarely observed in mice. It is plausible that this is caused by a disturbed shedding of key receptors involved in angiogenesis, such as vascular endothelial growth factor receptor 2 (VEGFR2) and Notch [74,182,183]. For example, ADAM10-mediated Notch cleavage is important for Notch activation which limits tip cell selection and sprouting formation in ECs [184]. Combined with the notion that interference with Notch signaling phenocopies vascular abnormalities that are observed in mice lacking ADAM10 [163,182,185], the reduced Notch signaling is most likely responsible for the observed pathological neovascularization in atherosclerotic lesions from mice lacking endothelial ADAM10. Mechanistically, it could be shown in vitro and in vivo that *Adam10* knockdown reduced the shedding of lectin-like oxidized LDL receptor-1 (LOX-1) and increased endothelial inflammation [63]. Interestingly, these effects are also completely opposite to the observed effects on atherosclerosis formation by endothelial *Adam17*-deficiency, as described above [63]. Although, as described, ADAM10 and ADAM17 share many substrates, they can also cleave various unique substrates. Based on the opposing effects in the endothelial-specific *Adam10* and *Adam17*-deficient studies and the notion that ADAM17 does not seem to be involved in LOX-1 cleavage [186], it is plausible that the underlying mechanism behind the pro-atherogenic effects of endothelial ADAM17 is due to the cleavage of an ADAM17-specific substrate, although this remains to be investigated.

Combined, it could be clearly demonstrated that ADAM10 plays a key role in atherosclerosis formation in a cell-specific manner. However, further studies are still needed to also explore the role of ADAM10 expressed in other cell types (such as SMCs) in atherosclerosis formation.

## 4. ADAMs in Cardiorenal Cross-Talk

As mentioned before, enhanced atherosclerosis is observed in animal models of CKD [187], and CKD patients have a significantly increased risk for CVD events and death [11,13]. However, the underlying mechanism of this cardiorenal cross-talk remains largely elusive. Uremic toxins accumulate in the circulation of CKD patients, as these cannot be sufficiently filtered, and thereby negatively influence the systemic vasculature, leading, for example, to protein/lipid modifications, immune cell activation, and endothelial dysfunction. Moreover, impaired kidney function augments hypertension and alters mineral bone metabolism, leading to vascular stiffness and calcification [22] (Figure 3). Furthermore, shedding and the release of various substrates into the circulation can be responsible for the cross-talk between the kidney and vasculature. Although ADAM10 and ADAM17 are clearly involved in both CKD and CVD, yet are more elaborately studied in the context of CVD, the role of ADAMs in the progression of renal disease and especially their role in cardiorenal disease is thus far a rather understudied field of research. Clear indications of the involvement of ADAMs in cardiorenal disease are for example provided by the NEFRONA study in which the circulating ADAMs were measured in 2570 CKD patients. The results indicated that soluble ADAMs (mainly ADAM17, but potentially also ADAM8, −9, or −10) are an independent risk factor for CV events in CKD patients [69,153]. Fascinatingly, vitamin D supplementation, which resulted in ADAM17 inhibition, prevented renal fibrotic and inflammatory lesions, which coincided with a decrease in systemic inflammation and related CV mortality [156].

In addition, certain shedding activities of ADAM10 and ADAM17 have been associated with CKD and the related CVD risk. For example, Klotho is cleaved in the kidney by both ADAM10 and ADAM17 [121] (Figure 2). Klotho is characterized as a vasculoprotective/anti-aging protein that is predominantly expressed in the distal tubule of the kidney where, interestingly, the highest ADAM10 and ADAM17 expression is also observed [59]. Klotho is known to interact with fibroblast growth factors 23 (FGF23). Captivatingly, high levels of FGF23 were observed in CKD patients, which coincided with a high expression of ADAM17. In turn, FGF23 is known to be an important indicator of oxidative stress and is associated with increased CV risk in CKD patients [188]. Additionally, soluble Klotho can exert protective effects against fibrosis and inflammation [189] in an FGF23-independent manner [190], thereby preventing CKD progression. Furthermore, Klotho is also expressed by ECs [60], which protects the vasculature from endothelial dysfunction [191]. In this manner, the shedding of Klotho by ADAM10/17 can exert not only effects on CKD but also on CKD-induced CVD [192].

Additionally, CXCL16 can also play a role in cardiorenal disease as it was observed that the expression of this chemokine and its receptor CXCR6 was significantly increased in the radial arteries of inflamed ESRD patients, characterized by high C-reactive protein levels compared to those in control non-inflamed ESRD patients [193]. Interestingly, these inflamed ESRD patients also had an increased ADAM10 expression in the radial arteries, which also coincided with an increased accumulation of foam cells in the vessel wall [193]. Combined, these results clearly suggest that ADAM10 can also play a key role in cardiorenal disease by influencing the CXCL16/CXCR6 pathway. It is likely that the release of EGFR ligands and other inflammatory molecules (IL-6R, TNF(R), etc.) in the kidney also impacts systemic vasculature [194,195].

Cross-talk within the kidney, particularly when damaged, has also been linked to extracellular vesicle (EV) release [196]. EVs are mediators of CVD [197] as well as systemic vascular calcification in CKD [198]. ADAMs can directly or indirectly modulate EV composition and function by their shedding activity and/or by being present themselves in EVs, affecting target cell responses such as endothelial (dys)function and VSMC calcification [199,200]. It has also been shown that EVs, more specifically microparticles, isolated from human atherosclerotic plaques carry active ADAM17 on their surface and that such microparticles can enhance the shedding of TNF and its receptor on ECs [85], indicating that ADAM17 can have systemic effects on inflammation. Furthermore, ADAM10 also has been observed to be present in EVs, although its exact effects are still elusive [100,199]. It is therefore likely that at least part of the cardiorenal communication is mediated by the release of soluble mediators, either involving ectodomains shed by the ADAMs or the release of EVs carrying ADAMs or their substrates.

## 5. Future Perspectives

Early detection and management of both CKD and CVD are crucial to prevent their progression and improve quality of life. Because it has been shown that ADAM10/17 influence many mediators of CKD-induced atherosclerosis, it would be very interesting to dive into the detailed mechanisms of action, as these so far remain rather elusive. ADAM10/17, to some extent, have cell-specific functions, future research should focus on elucidating these cell-specific effects of ADAM10/17 in CKD and especially CKD-induced atherosclerosis mouse models. Some suggestions to study cell-specific effects that are interesting in this context are, for example, EC (*Bmx* or *Tie*-Cre driven), podocyte (podocin (*Nphs2*)-Cre driven), or kidney epithelial cell (kidney-specific-protein (*Ksp*)-Cre driven) knockouts of *Adam10/17*. Although the first two suggested mouse models are not kidney-specific knockouts, this shortcoming can be overcome by transplanting a kidney from a knockout mouse into a wild-type recipient. These studies may contribute to understanding the role of these ADAMs in CKD-induced atherosclerosis and potentially open new doors for ADAM-based therapeutic approaches. This is crucial as, currently, treatment options to reduce CV mortality in patients with CKD are very limited, and clinical trials designed to reduce CV mortality in CKD patients have not proven to be successful yet. Notably, an exception is a clinical trial in which a statin and ezetimibe were given to non-dialysis patients with CKD [20,201,202], resulting in a 27% decrease in severe atherosclerosis-related events, although the benefits were rather limited as the treatment had no impact on mortality [203]. Therefore, new insights and treatment options are necessary to fight CKD-induced CVD morbidity and mortality.

A cell-specific targeting approach can be mediated via inactive rhomboid-like protease (iRhom2), which is involved in ADAM17 maturation and is only expressed in leukocytes [204,205]. A recent study by Hannemann et al. performed an investigation of atherosclerosis using mice that lack *iRhom2* and thereby have increased ADAM17 activation in leukocytes [206]. It was shown that the activation of ADAM17 resulted in a significantly decreased atherosclerotic lesion size, which is in line with the results observed in the previously discussed myeloid-specific *Adam17*-deficient model. Therefore, it would be very interesting and worthwhile to investigate more and other cell-/substrate-specific targeting approaches for ADAM10/17 to create ADAM10/17-based therapies in the context of (CKD-induced) atherosclerosis.

## Figures and Tables

**Figure 1 ijms-24-07309-f001:**
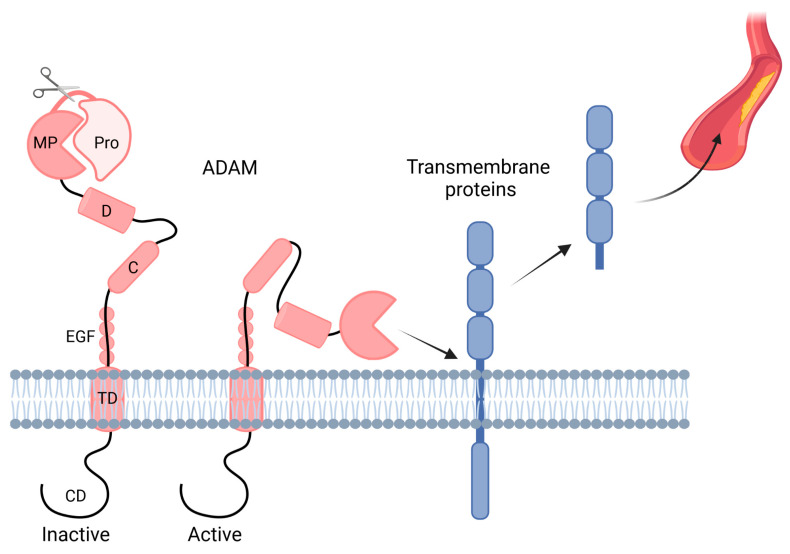
**ADAM domain structure.** A disintegrin and metalloproteinases (ADAM) present on the transmembrane consists of seven domains, namely the prodomain (pro), metalloprotease domain (MP), disintegrin domain (D), cysteine-rich domain (C), EGF-like domain (EGF; except for ADAM10 and ADAM17), transmembrane domain (TD), and cytoplasmic domain (CD). ADAMs are activated after the prodomain (Pro) is cleaved off, after which they can shed, among others, transmembrane proteins, which are released into the circulation. Created with BioRender.com.

**Figure 2 ijms-24-07309-f002:**
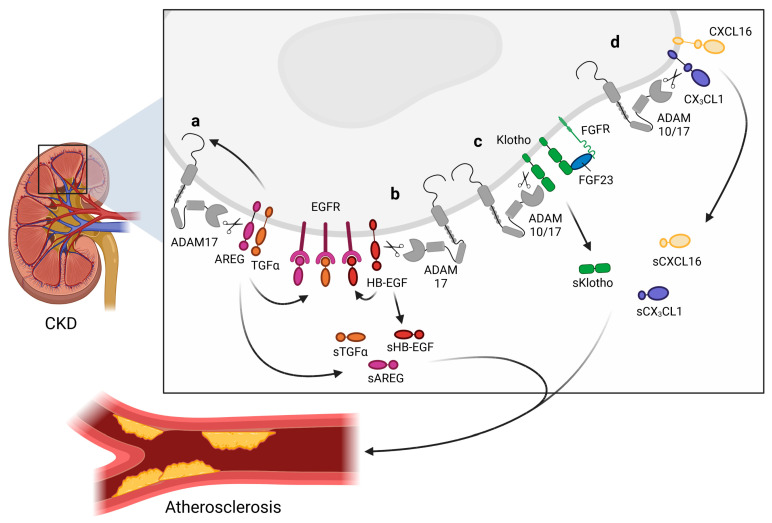
**Overview of shedding activities of a disintegrin and metalloproteinases (ADAM**) **10 and ADAM17 in the kidney.** The shedding of ADAM10 and ADAM17 substrates in chronic kidney disease (CKD). (**a**) Activation of the epidermal growth factor receptor (EGFR) pathway by shedding amphiregulin (AREG) and transforming growth factor α (TGFα) by ADAM17. AREG and TGFα shedding results in the upregulation of ADAM17 on the transmembrane. (**b**) Additionally, heparin-binding EGF-like growth factor (HB-EGF) is cleaved by ADAM10, also activating the EGFR pathway. (**c**) Shedding of Klotho is mediated by ADAM10 as well as ADAM17, resulting in soluble Klotho (sKlotho), which exerts renoprotective effects. Membrane-bound Klotho, fibroblast growth factor receptor (FGFR), and fibroblast growth factors 23 (FGF23) are from a trimeric signaling complex regulating kidney homeostasis. (**d**) ADAM10 and ADAM17 can shed chemokine (C-X3-C motif) ligand 1 (CX_3_CL1) and chemokine (C-X-C motif) ligand 16 (CXCL16), producing soluble chemokines. sAREG, sTGFα, sHB-EGF, sKlotho, sCX_3_CL1, and sCXCL16 are all released into the circulation and can translocate to the arterial endothelium and affect atherosclerotic processes. Created with BioRender.com.

**Figure 3 ijms-24-07309-f003:**
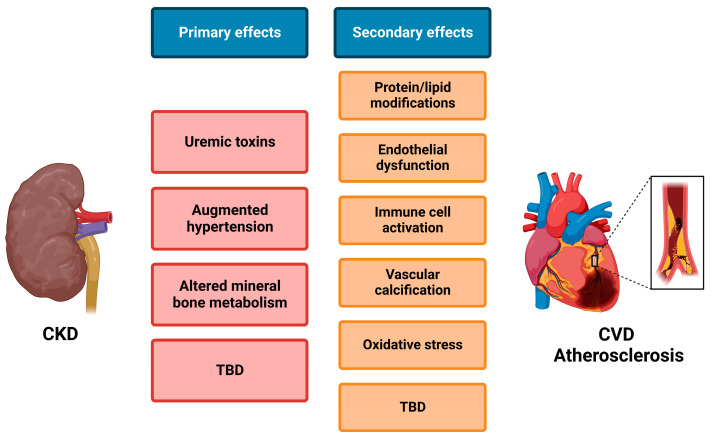
**Simplified overview of mechanisms involved in cardiorenal cross-talk.** Upon kidney damage and development of chronic kidney disease (CKD), several primary effects occur that can either directly or via subsequent secondary effects influence atherosclerosis and thus cardiovascular disease (CVD). Although several mechanisms are already known, many more remain to be defined (TBD). Created with BioRender.com.

**Table 1 ijms-24-07309-t001:** **Major risk factors specific for CKD, atherosclerosis, and CKD-induced atherosclerosis** [18,19,20,21,22].

General Risk Factors for CKD	General Risk Factors for Atherosclerosis	Risk Factors for CKD-Induced Atherosclerosis
Age	Age	Age
Diabetes mellitus	Diabetes mellitus	Diabetes mellitus
Hypertension	Hypertension	Augmented hypertension
	Dyslipidemia	Abnormal lipid and protein modifications
	Male gender	Uremic toxins
	Physical inactivity	Altered mineral bone metabolism
	Smoking	Oxidative stress
	Stress	
	Systemic low-grade inflammation	

Abbreviation: CKD—chronic kidney disease.

**Table 2 ijms-24-07309-t002:** **List of ADAM10 and/or ADAM17 substrates with relevance for kidney disease and atherosclerosis**.

ADAM	Inflammatory Mediators	Cell Process in Kidney Function/Disease	Reference
ADAM10	Betacellulin	Differentiation, fibrosis, migration, permeability, proliferation	[64,65,66]
	CADM1	Adhesion, apoptosis	[67,68,69]
	Corin	Blood pressure	[70]
	E-cadherin	Cell adhesion, wnt-signaling	[71,72]
	VE-cadherin	Endothelial permeability, angiogenesis	[73,74]
ADAM17	ACE2	Endothelial dysfunction, inflammation, oxidative stress	[75,76,77,78,79,80]
	AREG	Inflammation, fibrosis	[61,64]
	CD163	Inflammation, macrophage activation	[81]
	CD40	Immune suppression	[82]
	cKit ligand	Angiogenesis	[83]
	EMMPRIN	Lymphocyte cycling	[84]
	EPCR	Anticoagulation	[85]
	Ephrin B4	Sprouting	[84]
	FLT3L	Lymphopoiesis, progenitor differentiation	[86]
	IL-1R2	Anti-inflammatory	[87]
	IGFR1	Cell survival	[84]
	Jagged 1	Cell differentiation	[88]
	L-selectin	Adhesion	[89]
	L1-CAM	Adhesion	[90]
	Nox4	Oxidative stress	[91]
	NRP-1	Angiogenesis	[92]
	PECAM-1	Adhesion	[84]
	Sema 4D	Platelet activation	[93]
	Tie2	Angiogenesis	[94]
	TGFα	Cell proliferation	[64,95]
	TNFR1	Inflammation	[87,96,97]
	TNRF2	Inflammation	[96,97]
	VCAM-1	Adhesion, inflammation	[98]
ADAM10,	CD44	Angiogenesis, inflammation, migration	[99,100]
ADAM17	CD74	Leukocyte migration	[101]
	CX3CL1	Adhesion, inflammation, transmigration	[102,103,104,105,106]
	CXCL16	Leukocyte adhesion, inflammation	[104,105,106,107,108,109,110]
	DLL1	Migration, proliferation	[111]
	HB-EGF	Angiogenesis, cell survival, fibrosis, proliferation	[84,95,112,113]
	IL-6R	Inflammation	[114,115,116]
	ICAM-1	Leukocyte recruitment, inflammation	[117,118]
	JAM-A	Angiogenesis, transmigration	[119]
	KIM-1	Efferocytosis	[120]
	Klotho	Calcification, fibrosis, vascular dysfunction	[121,122,123,124,125,126,127,128]
	M-CSFR	Inflammation	[129]
	Meprin	Apoptosis, fibrosis, necrosis	[130,131,132]
	Notch	Angiogenesis, differentiation, EMT, fibrosis, inflammation	[133,134,135,136,137,138,139]
	Syndecan-1	Adhesion, migration, proliferation	[140,141,142,143,144]
	Syndecan-4	Adhesion, migration, proliferation	[143,145]
	TNFα	Fibrosis, inflammation	[61,97,146,147,148]
	TRANCE (RANKL)	Calcification, survival/proliferation, osteogenic SMC differentiation	[149,150,151]
	VEGFR2	Angiogenesis, cell survival, homeostasis, proliferation	[74,92]

Abbreviations: ADAM—a disintegrin and metalloproteinases; ACE2—angiotensin-converting enzyme 2; AREG—amphiregulin; CADM1—cell Adhesion Molecule 1; CX3CL1—chemokine (C-X3-C motif) ligand 1; CXCL16—chemokine (C-X-C motif) ligand 16; DLL1—Delta-like canonical Notch ligand 1; E-cadherin—epithelial cadherin; EGF—epidermal growth factor; EMMPRIN—extracellular matrix metalloproteinase inducer; EMT—epithelial-to-mesenchymal transition; EPCR—endothelial protein C receptor; FLT3L—Fms-related tyrosine kinase 3 ligand; HB-EGF—heparin-binding EGF-like growth factor; ICAM-1—intercellular adhesion molecule 1; IGFR1—insulin-like growth factor 1; IL-1RII—interleukin-1 receptor II; IL-6R—interleukin 6 receptor; JAM-A—junctional adhesion molecule A; KIM-1—kidney injury molecule-1; L1-CAM—L1 cell adhesion molecule; M-CSFR—macrophage colony-stimulating factor receptor; Nox4—NADPH oxidase 4; NRP-1—neuropilin-1; PECAM-1—platelet endothelial cell adhesion molecule-1; RANKL—receptor activator of nuclear factor κB ligand; Sema 4D—semaphorin 4D; SMC—smooth muscle cell; TGFα—transforming growth factor α; Tie2—tyrosine kinase with immunoglobulin-like loops and epidermal growth factor homology domains-2; TNFR1/II—tumor necrosis factor receptor 1/II; TNFα—tumor necrosis factor α; TRANCE—TNF-related activation-induced cytokine; VCAM-1—vascular cell adhesion protein 1; VE-cadherin—vascular endothelial cadherin; VEGFR2—vascular endothelial growth factor receptor 2.

## Data Availability

Not applicable.

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
