# Peer review of "ADAM10 and ADAM17, Major Regulators of Chronic Kidney Disease Induced Atherosclerosis?"

_ijms, 2023, doi:10.3390/ijms24087309_

Round 1

Reviewer 1 Report

The review paper is fine.

Author Response

Reviewer 1

The review paper is fine.

We would like to thank the reviewer for his/her positive evaluation of our review.

Reviewer 2 Report

To the authors,

The manuscript comprehensively summarizes the plausible role of ADAM10/17 in context of CKD and CVD along with possible avenues for therapeutic intervention in the context of CKD and CVD combined. It has been well written and needs minor grammatical checks.

Author Response

Reviewer 2

The manuscript comprehensively summarizes the plausible role of ADAM10/17 in context of CKD and CVD along with possible avenues for therapeutic intervention in the context of CKD and CVD combined. It has been well written and needs minor grammatical checks.

We have performed a thorough grammatical check and have added changes with track changes in the revised manuscript.

Reviewer 3 Report

Excellent and up-to-date review on very important and hot topic of interrelation between CKD and CVD and the molecular mediators of CVD in patients with CKD. Review brings new information to be translated (possibly in somewhat simplified way) also to clinical nephrologists and cardiologists with very important potential clinical implications including new modes of treatment.

1.       I would suggest to add another figure related to the putative mechanisms of CVD in CKD.

2.       How is the activity of ADAM10/17 related to renal function, proteinuria and kidney histology, e.g. degree of intestitial fibrosis, tubular atrophy, arteriolosclerosis, glomerulosclerosis, etc.?

3.       Is there any interaction between ADAM10/17 and MMPs (e.g. through TIMP-1, or TIMP-2)?

Minor comments:

1.       Line 144 „tubuli“ instead of „tubili“

2.       Line 165 „transmembrane proteins“ instead of „transmembrane protein“

Author Response

Reviewer 3

Excellent and up-to-date review on very important and hot topic of interrelation between CKD and CVD and the molecular mediators of CVD in patients with CKD. Review brings new information to be translated (possibly in somewhat simplified way) also to clinical nephrologists and cardiologists with very important potential clinical implications including new modes of treatment.

  1. I would suggest to add another figure related to the putative mechanisms of CVD in CKD.

As suggested by the reviewer we have added another figure in the revised manuscript (Figure 3).

  1. How is the activity of ADAM10/17 related to renal function, proteinuria and kidney histology, e.g. degree of intestitial fibrosis, tubular atrophy, arteriolosclerosis, glomerulosclerosis, etc.?

As suggested we have included a section about the effect of ADAM10/17 on renal function and pathology in lines 304-315:

“Besides the describes effects of on CKD, the shedding of a variety of transmembrane proteins have also been shown to influence the morphology of the kidney as for example cleavage of CXCL16 by ADAM10 led to lupus nephritis and acute tubular necrosis [107, 108], while IL6-R shedding by ADAM10/17 resulted in acute crescentic glomerulonephritis, lupus nephritis [114] and Notch shedding by ADAM10/17 to renal fibrosis and glomerulosclerosis [164, 168]. Additionally, the shedding of TNF-α and TGF-β by ADAM17, resulted in increased fibrosis, glomerulosclerosis, inflammation, protein matrix accumulation, and neutrophil and macrophages infiltration by both the TNF receptor (TNFR) and EGF receptor (EGFR) signaling pathways [47, 61, 147]. Furthermore, it has been shown that substrates of ADAM10/17, like Meprin influence kidney function as the Meprin inhibitor, actinonin lowered blood urea nitrogen and serum creatinine levels in the presence of renal sepsis [131].“

  1. Is there any interaction between ADAM10/17 and MMPs (e.g. through TIMP-1, or TIMP-2)?

In addition to the section in line 130-131:

‘Additionally, ADAMs expression is regulated by endogenous proteins known as tissue inhibitors of metalloproteases (TIMPs)’.

We have now added the following information in lines 132-134:

TIMPs are well known to also regulate matrix metalloproteases (MMPs) [53]. Interestingly, there seems to be a link between ADAMs and MMPs as for example ADAM-mediated release of TNF-α induces MMPs (MMP-2, -8, -9) via a positive feedback loop [54-56].”

Reviewer 4 Report

This review considers up to date literature data of the role of A Disintegrin and Metalloproteases ADAM 10 and ADAM 17 in chronic kidney disease and cardiovascular disease as well as their role in CKD induced CVD.

This is well-down, systematic and extensive review with 201 cited references and can be very interesting and inspiring for scientists who are involved in CKD research.

I have only one suggestion for the authors. It may be appropriate to include Method section, where you would introduce the search process, including Database used, keywords, and inclusion and exclusion criteria for articles, which were used while preparing this manuscript.

Author Response

Reviewer 4

This review considers up to date literature data of the role of A Disintegrin and Metalloproteases ADAM 10 and ADAM 17 in chronic kidney disease and cardiovascular disease as well as their role in CKD induced CVD.

This is well-down, systematic and extensive review with 201 cited references and can be very interesting and inspiring for scientists who are involved in CKD research.

I have only one suggestion for the authors. It may be appropriate to include Method section, where you would introduce the search process, including Database used, keywords, and inclusion and exclusion criteria for articles, which were used while preparing this manuscript.

We would like to thank the reviewer for his/her positive evaluation of our review. This review is a narrative review, rather than a systematical one. Therefore, we have not elaborated on methodological details in the text. In order to make this more clear we have stated in the abstract and introduction that this is a “narrative” review.

As reviewer-only information, we would like to elaborate on your PubMed search criteria that we have used:

ADAM AND CKD

ADAM10 AND CKD

ADAM17 AND CKD

ADAM AND Kidney

ADAM10 AND Kidney

ADAM17 AND Kidney

ADAM AND CKD OR chronic kidney disease

ADAM10 AND CKD OR chronic kidney disease

ADAM17 AND CKD OR chronic kidney disease

ADAM AND CVD OR cardiovascular disease

ADAM10 AND CVD OR cardiovascular disease

ADAM17 AND CVD OR cardiovascular disease

ADAM AND Atherosclerosis

ADAM10 AND Atherosclerosis

ADAM17 AND Atherosclerosis

ADAM AND CKD AND CVD OR Cardiovascular disease

ADAM10 AND CKD AND CVD OR Cardiovascular disease

ADAM17 AND CKD AND CVD OR Cardiovascular disease

ADAM AND CKD AND Atherosclerosis

ADAM10 AND CKD AND Atherosclerosis

ADAM17 AND CKD AND Atherosclerosis

In order to keep the review comprehensible, we have focused on recent manuscripts specifically about CKD (no other kidney pathologies), atherosclerosis (no other CVDs) and only ADAM10 and ADAM17.